# Reasonable Coal Pillar Width and Surrounding Rock Control of Gob-Side Entry Driving in Inclined Short-Distance Coal Seams

Fulian He [1,2], Wenli Zhai [1,*], Jiayu Song [1], Xuhui Xu [1], Deqiu Wang [1] and Yanhao Wu [1]

1   School of Energy and Mining Engineering, China University of Mining and Technology (Beijing), Beijing 100083, China; flhe1966@126.com (F.H.); bqt2100101001t@student.cumtb.edu.cn (J.S.)
2   Beijing Key Laboratory for Precise Mining of Intergrown Energy and Resources, China University of Mining and Technology (Beijing), Beijing 100083, China
*   Correspondence: wlzhai8@163.com

**Abstract:** During gob-side entry driving under complex conditions in inclined short-distance coal seams, the roadway loses stability and deforms seriously, which affects the safety and efficiency of mine production. In this study, a reasonable coal pillar width was explored by means of on-site investigation, theoretical analysis, numerical simulation, and engineering tests. The following research results were obtained: (1) In selecting a reasonable coal pillar width, the influences of the position of residual coal pillars, stratum spacing, main roof breakage, roadway section in the upper coal seam should be considered. From established mechanical models of inclined gob-side roadways, the maximum floor failure depth is 27 m and the concentrated influence range of the #1 coal pillars is 11 m. (2) The stress states of coal pillars with different widths were analyzed by numerical simulation. As the coal pillar width increases, the peak value of the stress increases first and then decreases. Based on the site geological conditions, the optimum coal pillar width was determined to be 8 m, which is consistent with the theoretical calculation results. (3) A new pressure-yield support technology was proposed, and its on-site application confirmed its notable roadway control effect. Our research can provide theoretical support for the control of roadways surrounding rock under similar engineering background conditions.

**Keywords:** inclined coal seams; gob-side entry driving; numerical simulation; reasonable coal pillar width; yield support

## 1. Introduction

As the depth of coal mining rises, an increasing number of coal mining areas are faced with the technical problem of short-distance multi-seam mining [1–4]. During the mining of short-distance coal seams, the gob and coal pillars left from the upper coal seams greatly affect the selection and control of the roadway location in the lower coal seams [5–7]. An unreasonable roadway location in the lower coal seams often causes extreme roadway deformation, which greatly affects normal recovery of the working face [8–10]. Therefore, gob-side entry driving in lower coal seams should not only consider factors influencing lower-coal-seam mining, but also avoid the influence of the high stress concentration of residual coal pillars in the upper coal seams, which is of great significance for safe and efficient coal mining [11].

Experts have conducted extensive research on the stress distribution characteristics of short-distance coal seams, the reasonable coal pillar width in gob-side entry driving, and the surrounding rock control [12–14]. Ref [15] analyzed the correlation between the deformation failure of bottom coal in the gob-side entry and coal pillar width by theoretical analysis, numerical calculation, and similar simulation experiments. Ref [16] studied the mechanical behavior of coal pillar load on the floor rock strata in multi-seam downward mining and discovered that stress was highly concentrated in the floor rock strata near

coal pillars, with the stress concentration degree in floor rock strata positively correlated with the distance between the surrounding rock and coal pillars. Quang et al. (2020) [17] adopted numerical simulation software and determined a reasonable design for roadway supporting pillars. Gao (2004) [18] used the method of similarity simulation experiments to study the laws of roof activity. Ghasemi et al. (2014) [19] applied fuzzy logic to predict sate pillar sizing in room and pillar coal mines. Hu et al. (2012) [20] summarized the main factors causing roadway damage in adjacent gobs and found that roadway deformation can be effectively controlled using high-preload and dense-bolt (cable) support systems. With respect to problems such as serious asymmetric deformation and support failure of gob-side roadways, He et al. (2020) [21] proposed a hydraulic fracture technique to pre-fracture the two hard roofs, and the validity of the above roof treatment was verified via field application. Meanwhile, some experts have improved the geological and mechanical conditions of the coal mining process through backfilling technology [22,23].

Based on the engineering background of the 4602 head roadway in the Honghui Coal Mine, this study establishes a theoretical model for floor failure in short-distance coal seams and analyzed the floor failure range of the residual coal pillars. In addition, it establishes a mechanical model of inclined gob-side roadways and deduces an expression for the influence range of low stress areas and its influencing factors. Moreover, the stress distribution characteristics of coal pillars with different coal pillar widths in gob-side entry driving were analyzed by numerical simulation; a reasonable coal pillar width was comprehensively determined, and the yield-anchor mesh-cable support system was proposed. The proposed support system effectively controls the integrity of the roadway surrounding rock and realizes the safety and stability of the roadway support.

## 2. Engineering Background

### 2.1. Engineering Profile

The #2 coal seam of the 4602 working face was mined with a dip length of 116 m, a design strike length of 915 m, and the following coal seam parameters: buried depth 354~284 m, average dip angle 17°, average thickness 3.0 m, firmness coefficient f 1.8~2.2, and medium-level bedding development. The average thickness of coal seam #1 is 8.8 m, the average dip angle of the coal seam is 15°, the width of the remaining coal pillar is 10 m, and the vertical distance from coal seam #2 is approximately 19 m. The shallow and deep parts of the 4602 working face are the 4502 gob and the 4702 gob, respectively. The 4602 working face is an island face whose location is illustrated in Figure 1. A mudstone false roof that is 1 m thick exists in the #2 coal seam roof. The immediate roof, whose thickness is unstable and averages 19 m, comprises fine sandstone and siltstone, with gray and argillaceous cementation and developed fractures. The immediate floor, whose average thickness is 2.45 m, comprises argillaceous siltstone with gray white and argillaceous cementation and poor hardness, and it softens when encountering water. The lithologies of the coal seam roof and floor are shown in Figure 2.

### 2.2. Originally Adopted Support Scheme and Deformation and Failure Characteristics

In the originally adopted gob-side entry driving, the roadway has a rectangular section with a net width of 4000 mm and a net height of 3000 mm. The roadway roof and sides adopt Φ 20 mm × 2400 mm high-strength bolt supports with a row spacing of 800 mm × 800 mm. The anchor cables are Φ 18.9 mm × 5000 mm mining anchor cables; each row has 3 cables with a spacing of 1200 mm, and the row spacing is 1600 mm, with the bolts and anchors fixed in sections, and resin cartridges were used. After the above support scheme is adopted, the roadway surrounding rock deforms markedly during recovery, especially the gob-side roof and sides of the roadway, where the bolts move outward with the deformation of the surrounding rock, with a maximum inward movement of 700 mm (Figure 3a). At the same time, local floor heave occurs in the roadway, with the bottom heave of the roadway floor occurring for a maximum of 800 mm (Figure 3b). The recovery requires repeated side expansion and floor reshaping. If the situation becomes too serious,

grouting and re-building metal U-shaped steel shed supports will be needed after the repair. In view of the above roadway deformation characteristics of gob-side entry driving, coal pillar width and support parameters require further optimization.

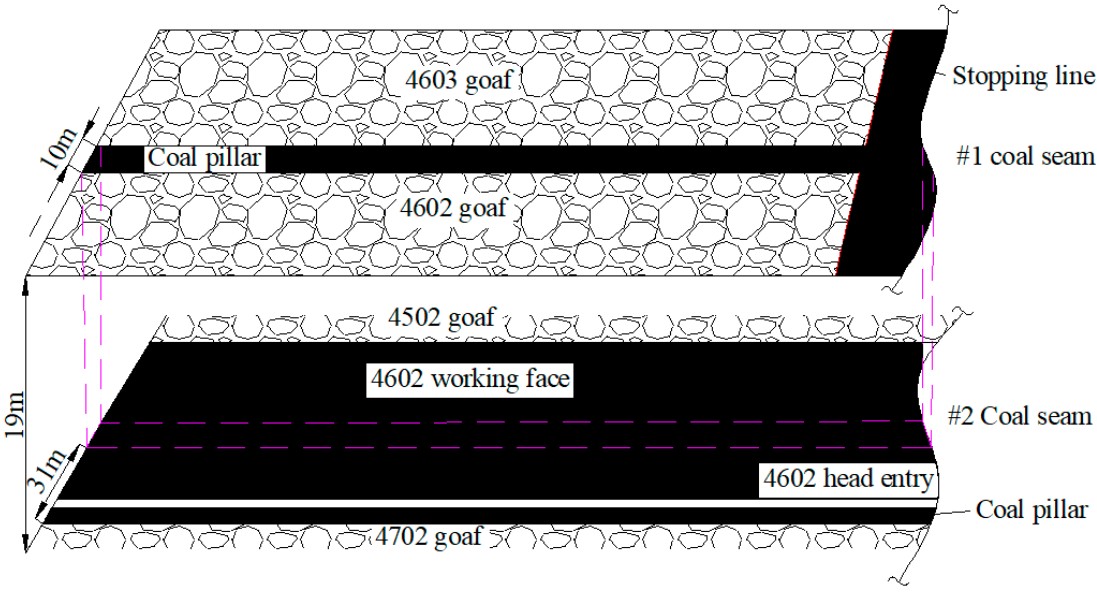

**Figure 1.** Spatial position diagram of the working face.

| Histogram | Rock stratum | Thickness/m | Lithology description |
|---|---|---|---|
| | coarse sandstone | 16.5 | gray white, coarse-grained structure, massive structure, mainly composed of quartz |
| | No.1 coal seam | 8.8 | black, massive, with pyrite film, Semi bright briquette |
| | siltstone | 12.0 | dark gray, dense, brittle, developed fissures |
| | fine sandstone | 6.0 | dark gray, dense and hard, containing a large number of muscovite fragments, mixed with coal line |
| | mudstone | 1.0 | grayish black, argillaceous cementation, softened in case of water |
| | No.2 coal seam | 3.0 | black, massive, with pyrite film, Semi-bright briquette |
| | pelitic siltstone | 2.45 | dark gray, dense and hard, argillaceous cementation, with coal line |
| | siltstone | 6.5 | dark gray, dense, brittle, developed fissures |

**Figure 2.** Rock stratum histogram of working face.

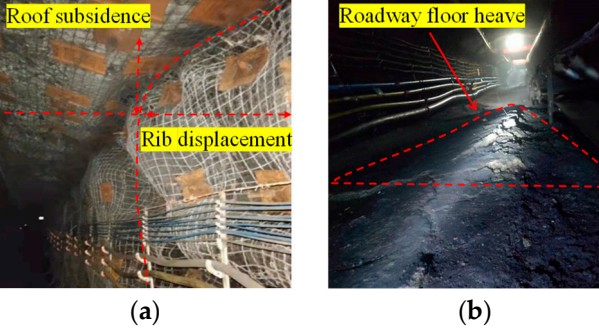

(**a**)                  (**b**)

**Figure 3.** Deformation and failure of a gob-side coal roadway. (**a**) Deformation of the roof and rip. (**b**) Roadway floor heave.

### 3. Calculation of Coal Pillar Width of Gob-Side Entry Driving

*3.1. Study on the Floor Failure Range of Residual Coal Pillars*

Residual coal pillars exist after mining the #1 coal seam. The floor failure depth resulting from the residual coal pillars can directly reveal whether the lower-coal-seam mining is affected. The coal pillar load acts on the floor and causes damage (Figure 4).

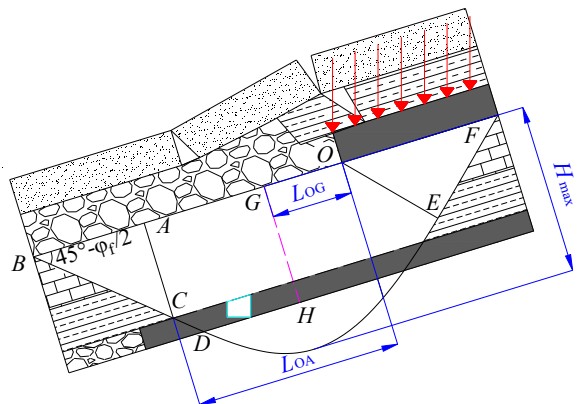

**Figure 4.** Floor failure model.

The *ED* curve is a logarithmic spiral, and its formula is (Xu et al., 2021 [24]):

$$r = r_0 e^{\alpha \cdot \tan \varphi_f} \tag{1}$$

where $r$ is the spiral radius with $O$ as the origin at an angle of $\alpha$ with $r_0$; $r_0$ is the length of $OE$; $\alpha$ is the angle between $r$ and $r_0$; and $\varphi_f$ is the internal friction angle of rock strata.

Since $\angle OFE = \angle EOF = \pi/4 + \varphi/2$,

$$r_0 = B/2 \cos\left(\pi/4 + \varphi_f/2\right) \tag{2}$$

where $B$ is the width of the coal pillar acting on the floor.

When the floor reaches the maximum yield failure depth, $\alpha$ can be calculated by:

$$\alpha = \pi/4 + \varphi_f/2 \tag{3}$$

Then the maximum floor failure depth can be expressed as:

$$H_{\max} = \frac{B \cos \varphi_f}{2 \cos\left(\frac{\pi}{4} + \frac{\varphi_f}{2}\right)} e^{\left(\frac{\pi}{4} + \frac{\varphi_f}{2}\right) \tan \varphi_f} \tag{4}$$

According to Equation (4), the maximum floor failure depth under the influence of residual coal pillars is related to the coal pillar width and floor lithology. Based on the actual geological conditions of the coal and the rock strata, the internal friction angle $\varphi_f$ is 38° and the width of the coal pillars $B$ is 10 m. They are substituted into Equation (4): $H$max = 27 m, which is larger than the average distance of 19 m between the two coal seams. Therefore, the failure range of residual coal pillars in the upper coal seams has a significant influence on the lower-coal-seam mining and the mining roadway layout.

Based on the above analysis, the #2 coal seam roadway can be arranged at the position where the residual coal pillar stress is less concentrated on the floor, which is conducive to roadway stability. The influence range of the high floor stress concentration of residual coal pillars in the #1 coal seam $L_{OG}$ is shown in Figure 4,

$$L_{OG} = (h_1 + h_2) \tan \varphi \tag{5}$$

where $h_1$ is the thickness between the #1 and #2 coal seams, m, calculated as 19 m; $h_2$ is the #2 coal seam thickness, m, calculated as 3 m; and $\varphi$ is the propagation angle of stress influence, calculated as 25°.

Based on the calculation of $L_{OG}$, that is, at the #1 coal seam floor, the concentrated influence range of the coal pillars is 11 m. With reference to the calculation results, if the #2 coal roadway is arranged over 11 m away from the residual coal pillars of the #1 coal seam, the roadway will be less affected by the residual coal pillars of the #1 coal seam.

### 3.2. Study on the Reasonable Coal Pillar Width

After the recovery of the working face in the upper section, the gob-edge immediate roof collapses, and the main roof rotates. The gangue supports the rock block $B$ of the main roof; the rock block $B$ and adjacent blocks squeeze each other to form a relatively stable structure. The fracture of the main roof divides the lateral abutment pressure into the obvious internal stress field $S_1$ (stress falling area) and the external stress field $S_2$ (stress rising area). The location of the main roof fracture is the boundary of the two stress fields (Figure 5a). To provide a good stress environment for the roadway and to facilitate roadway maintenance, roadways are generally arranged in the stress falling area in coal pillar width selection (Song et al., 1986 [25]).

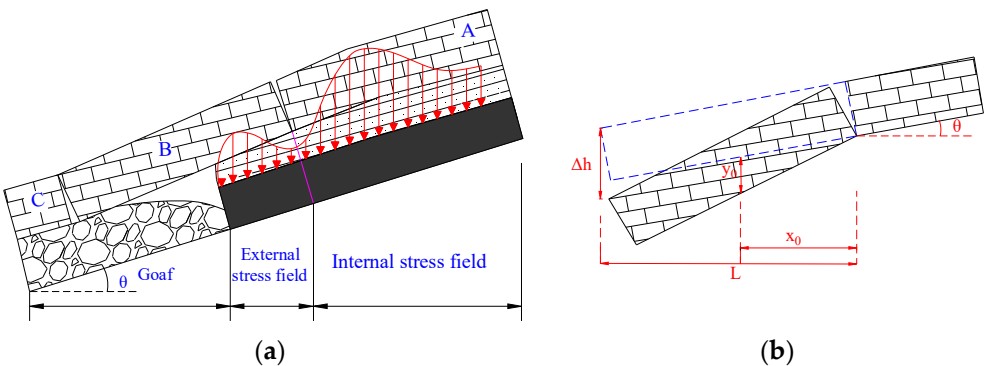

(**a**)  (**b**)

**Figure 5.** Mechanical model. (**a**) Distribution diagram of the internal and external stress fields. (**b**) Breaking structure model of the main roof.

In the internal stress field, the abutment pressure $x$ m away from the coal wall can be expressed as:

$$\sigma_x = G_x y_x \tag{6}$$

where $G_x$ is the stiffness of the coal body $x$ m away from the coal wall, Pa; $y_x$ is the deformation of the coal body $x$ away from the coal wall, m.

As the abutment pressure develops into the deep part of the coal wall, the stiffness of the coal body gradually rises while the compression gradually decreases. This is simplified as a process of linear variation. According to the geometric relationship, then:

$$y_x = \frac{y_0}{x_0}(x_0 - x), G_x = \frac{G_0}{x_0}x \tag{7}$$

where $G_0$ is the maximum stiffness of the coal body within the internal stress field, Pa; $y_0$ is the compression of the gob-edge coal body, m; $x_0$ is the distribution range of the internal stress field, m. According to Equations (6) and (7), the abutment pressure of the internal stress field is

$$\int_0^{x_0} \sigma_v dx = \frac{G_0 y_0 x_0}{6} \tag{8}$$

According to the theory of internal and external stress fields, the abutment pressure within the internal stress field on the coal body around the stope is equal to the self-weight of the main roof strata during the first weighting of the working face; thus,

$$\frac{G_0 y_0 x_0}{6} = abC_0\gamma \tag{9}$$

where $a$ is the inclined length of the working face, m; $C_0$ is the first weighting step distance of the adjacent working face, m; $b$ is the thickness of the main roof strata, m; and $\gamma$ is the bulk density of the main roof rock, kN/m$^3$. According to Figure 5b, then:

$$\frac{y_0}{x_0} = \frac{\Delta h}{L} = \frac{M - h(K_c - 1)}{L} \tag{10}$$

$$y_0 = \frac{x_0}{L}[M - h(K_c - 1)] \tag{11}$$

where $M$ is the mining thickness of the coal seam, m; $h$ is the thickness of the immediate roof rock strata, m; $\Delta h$ is the maximum subsidence of the key block, m; $L$ is the span of the hinged rock beams, which is nearly equal to the periodic weighting step distance of the fully mechanized caving face, m; and $K_c$ is the dilatancy coefficient of the immediate roof rock.

According to the inclusion theory, the stiffness of the coal body $G_0$ (Zhang et al., 2016 [26]) in the plastic state is

$$G_0 = \frac{E}{2(1 + v)\xi} \tag{12}$$

where $E$ is the elastic modulus of the coal body, *Pa*; $v$ is Poisson's ratio; and $\xi$ is the influence coefficient, which is related to the development of fractures in the coal body.

The range of the internal stress field $x_0$ can be obtained based on Equations (9)~(12):

$$x_0 = \sqrt{\frac{12abC_0\gamma\xi L(1 + v)}{E[M - h(K_c - 1)]}} \tag{13}$$

According to geological data from the mine and on-site measurements, $a = 120$ m, $b = 12$ m, $C_0 = 35$ m, $\gamma = 27$ kN/m$^3$, $\xi = 0.85$, $L = 16$ m, $v = 0.3$, $E = 24$ GPa, $M = 3$ m, $h = 7$ m, and $K_c = 1.32$. These values are substituted in Equation (13) to obtain the internal field range $x_0 = 12.58$ m. The constraints of the reserved coal pillars in gob-side entry driving is to arrange roadways within the internal stress field. The 4602 head roadway is designed 4.6 m wide, and the width of coal pillars should not exceed 8 m.

A reasonable width should ensure not only the stability of the roadway surrounding the rock during excavation, but also the safety and stability of the working face during recovery. Although the 4602 head roadway can be arranged in a low stress environment by reserving 3~5 m small coal pillars, the 4602 working face belongs to an island face which is also affected by the residual coal from the #1 coal seam. Hence, the 4602 working face

is likely to experience large-scale instability and damage. Internal fractures of coal pillars may develop, leading to poor stability and air leakage in the gob, unfavorable for fire prevention. Meanwhile, according to the above calculation, the roadway should also be arranged outside the influence range of the coal pillar stress concentration in the upper coal seam, so the coal pillar width is preliminarily determined to be in the range of 6~8 m. Due to the limitations of theoretical model calculation, the final coal pillar width is determined by calculating the stress distribution characteristics of coal pillars with different widths through numerical simulation.

## 4. Numerical Simulation of Coal Pillar Width of Gob-Side Entry Driving in Inclined Coal Seams

### 4.1. Establishment of the Numerical Model

According to the mining relationship and the roof and floor strata conditions of the 4602 working face, a numerical calculation model that is 470 m long, 200 m wide, and 200 m high is established (Figure 6). In order to simulate the overburden load, the upper boundary stress of the model is set as 7.5 MPa. The displacement on the left, right, and bottom boundaries is fixed to 0. The lateral pressure coefficient is set as 1.2, and the coal and rock masses are calculated by the Mohr–Coulomb model. The horizontal distance between the #1 coal seam residual pillars and the #2 coal seam is 31 m; therefore, the above calculated stress concentration influence range of residual pillars is 11 m and the range of the internal stress field is 8 m. The reasonable pillar width can be further determined by simulating the stress distribution characteristics of the roadway surrounding the rock with pillar widths of 4 m, 6 m, 8 m, 10 m, 12 m, and 14 m, respectively. The calculation process is as follows: model establishment → original rock stress equilibrium → excavation of the #1 coal → excavation of the 4502 and 4702 working faces in the #2 coal seam → excavation of the 4602 head roadway → calculation result analysis. The mechanical parameters of the coal and rock mass in numerical simulation are obtained by conversion on the basis of laboratory test data. Table 1 gives the mechanical parameters of the coal rock mass.

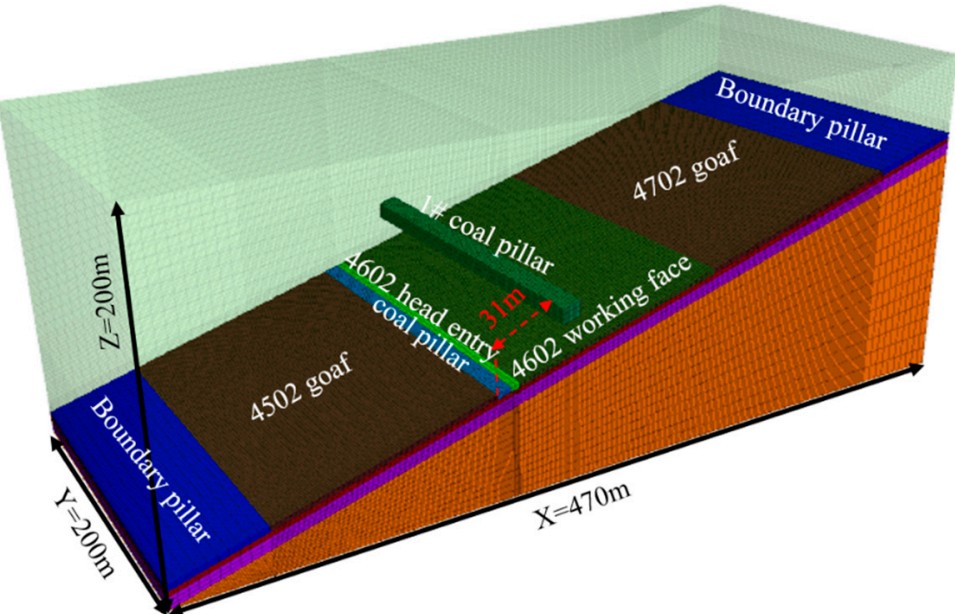

**Figure 6.** Three-dimensional numerical model.

**Table 1.** Mechanical parameters of the coal rock mass.

| Lithology | $\rho$ (kg/m$^3$) | $E_i$ (GPa) | $c_i$ (MPa) | $\varphi_i$ (deg.) | $\sigma_{ti}$ (MPa) | $v_i$ |
|---|---|---|---|---|---|---|
| Coarse sandstone | 2556 | 10.25 | 7.64 | 29.31 | 1.86 | 0.20 |
| No.1 coal seam | 1335 | 2.46 | 2.91 | 23.59 | 1.35 | 0.25 |
| Siltstone | 2548 | 7.25 | 4.59 | 28.42 | 1.62 | 0.18 |
| Fine sandstone | 2552 | 10.63 | 6.90 | 31.74 | 1.72 | 0.19 |
| Mudstone | 1220 | 3.58 | 1.50 | 15.53 | 0.80 | 0.27 |
| No.2 coal seam | 1329 | 2.97 | 2.35 | 21.86 | 1.10 | 0.23 |

### *4.2. Analysis of the Model Results*

The analysis of the influence range of the residual coal pillar can provide a certain basis for the selection of the width of the coal pillar during the mining of the lower coal seam. From the numerical simulation results (Figure 7), it can be seen that the peak value of the advance abutment pressure during the mining of the #2 coal seam is 16.9 MPa. The residual coal pillar has a certain impact on the mining of the #2 coal seam, and the influence range of the stress concentration is 10.5 m (Figure 7b), which is almost identical to the theoretical calculated value of 11 m for the concentrated influence range of the remaining coal pillar of the coal seam; thus, the accuracy of theoretical calculation is further verified. Therefore, when roadways are driven along the goaf in coal seam #2, the roadways should be arranged to avoid the influence of the #1 residual coal pillar as much as possible.

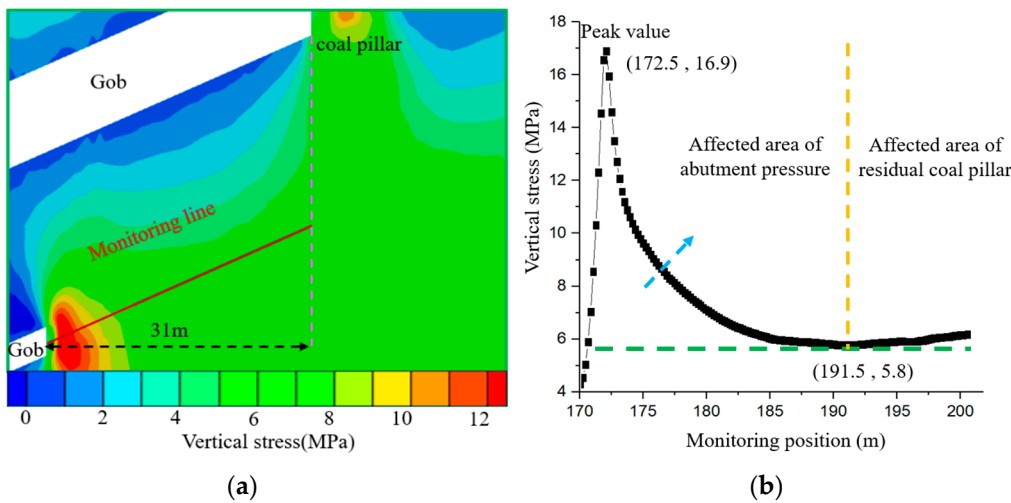

**Figure 7.** Influence range of residual coal pillar stress. (**a**) Stress distribution. (**b**) Stress monitoring curve.

When the coal and rock masses boast a strong bearing capacity, stress concentration is likely to occur in a certain range. Stress is re-distributed after roadway excavation. The simulation and analysis of the stress distribution in the coal pillars is of certain guiding significance for determining the reasonable coal pillar width. The stress nephograms and distribution curves of different coal pillar widths were obtained by numerical calculation results (Figure 8).

According to Figure 8, overall, the stress distributions of different coal pillar widths show the following rules:

(1) At coal pillar widths of 8~14 m, the internal stress of coal pillars presents an asymmetric bimodal distribution with high stress on the gob sides and low stress on the roadway sides. When the width is below 6 m, the internal stress presents a single peak distribution.

(2) After the roadway excavation, the vertical stress is re-distributed. At a small coal pillar width, stress is concentrated on the right side of the roadway. As the width increases, the position of the stress concentration gradually shifts to the gob side.

(3) At widths of 4 m and 6 m, fractures develop in the coal pillars, leading to serious damage and poor bearing capacity, which fails to form stress concentration. However, as the width continuously increases to 8 m, the mechanical properties of the coal and rock mass improve, and the peak value of the stress nucleus rises. When the width reaches 10 m, the maximum stress peak value is 11.30 MPa. When the width exceeds a certain value and reaches 12 m, the abutment pressure increases and surpasses the ultimate strength of the coal body. In this case, the stress in the coal pillar is relieved, and the peak value of the stress nucleus decreases. The peak stress value rises first and then decreases in turn in the following order: 1.67 MPa, 7.22 MPa, 10.67 MPa, 11.30 MPa, 10.27 MPa, and 10.25 MPa.

(4) When the width of the coal pillar reaches 8 m, the coal pillar boasts an overall good bearing capacity. An increase in the width improves the integrity of the coal pillar, but this will cause a significant waste of coal resources. However, a width below 8 m corresponds to poor integrity and difficult maintenance; at the same time, it is easy to cause air leakage in the coal pillar, which is not conducive to fire prevention management in the goaf. Therefore, a width of 8 m can ensure the stability of coal pillars through appropriate measures and maximize the recovery of coal resources.

In summary, when the width of the coal pillar is 8 m, coal pillars have an overall good bearing capacity. Meanwhile, the roadway is within the internal stress field, and can also avoid the impact of high stress caused by the residual coal pillars in the upper coal seams. Under strengthened support, the roadway can fully meet the requirements for basic transportation, ventilation, and side walking during recovery. Moreover, strengthened support can maximize the recovery of coal resources and ensure the safety and stability of roadway during recovery. As a result, it is determined that the coal pillar width of gob-side entry driving should be set to 8 m through theoretical analysis and on-site actual situation.

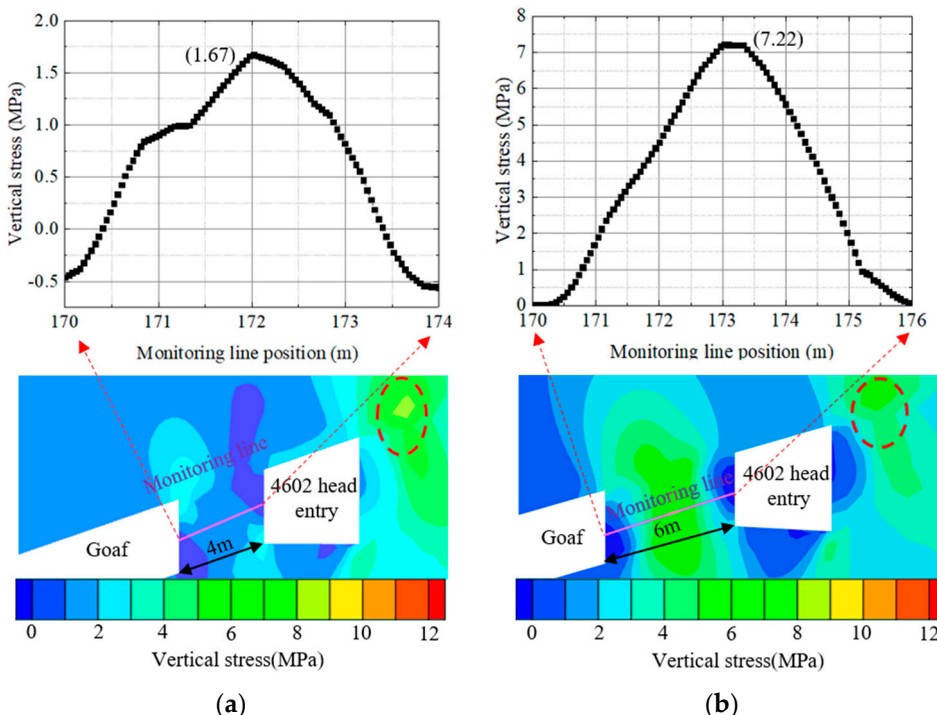

(**a**)　　　　　　　　　　　　　　　　　(**b**)

**Figure 8.** *Cont.*

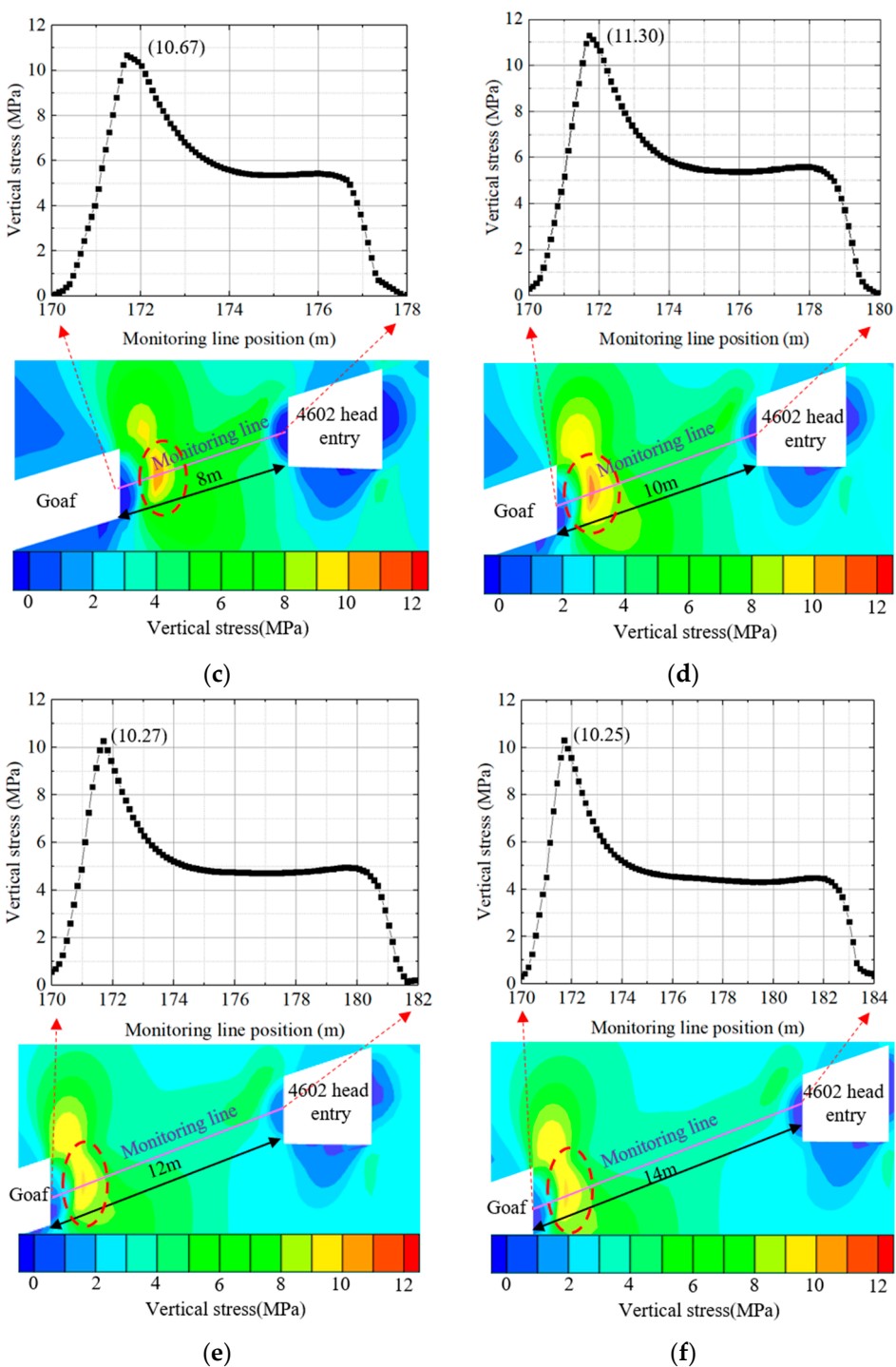

**Figure 8.** Stress distribution of different coal pillar widths. (**a**) Stress variation of a 4 m coal pillar. (**b**) Stress variation of a 6 m coal pillar. (**c**) Stress variation of an 8 m coal pillar. (**d**) Stress variation of a 10 m coal pillar. (**e**) Stress variation of a 12 m coal pillar. (**f**) Stress variation of a 14 m coal pillar.

## 5. Determination of the 4602 Head Roadway Support Scheme

The 4602 head roadway is in a high stress environment for gob-side entry driving. Hence, targeted design and improvements in anchor net cable support are required. The stability of a roadway surrounding rock can be controlled using yield support technology [27–31]. The principle of "yield before resist, a combination of yield and resist" was adopted to control the deformation of the roadway surrounding rock. "Pressure yielding" by the use of pressure-yield bolts can lead to a certain plastic deformation of roadways

surrounding rock, so that the deformation of surrounding rock can be released to a certain extent. "Pressure resisting" by high-strength bolts can resolutely resist the deformation of roadways surrounding rock. "Pressure resisting" does not mean simply improving the bolt strength. When the bolt strength increases to a certain extent, the mechanical parameter ratio of the roadway surrounding rock varies slightly. Meanwhile, when the bolt has a higher strength, its elongation is lower, and fails to resist the deformation of the roadway surrounding rock. At this time, the performance of the pressure-yield bolts can be brought into full play.

### 5.1. Bolt Support System

The parameters of the support system are as follows: Φ 20 mm × 2600 mm high-strength anchor bolts, 150 mm × 150 mm × 10 mm high-strength trays with 260 N·m tightening torque. The anchor bolts are matched with pressure-yield pipes. The pressure-yield distance is 26 mm and the load-yield point is 120~150 kN (Figure 9a). The anchor bolt layout is shown in Figure 10, and surface support is realized by a prepared metal mesh.

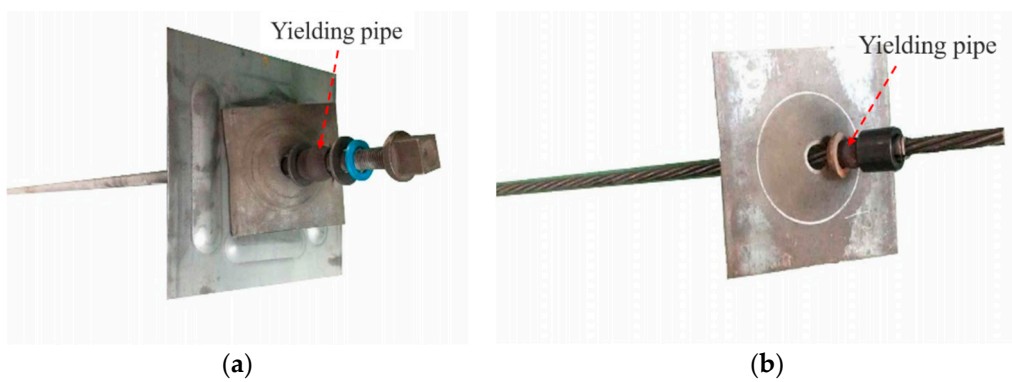

(a)          (b)

**Figure 9.** Pressure-yield supporting materials. (**a**) Pressure-yield anchor bolt. (**b**) Pressure-yield anchor cable.

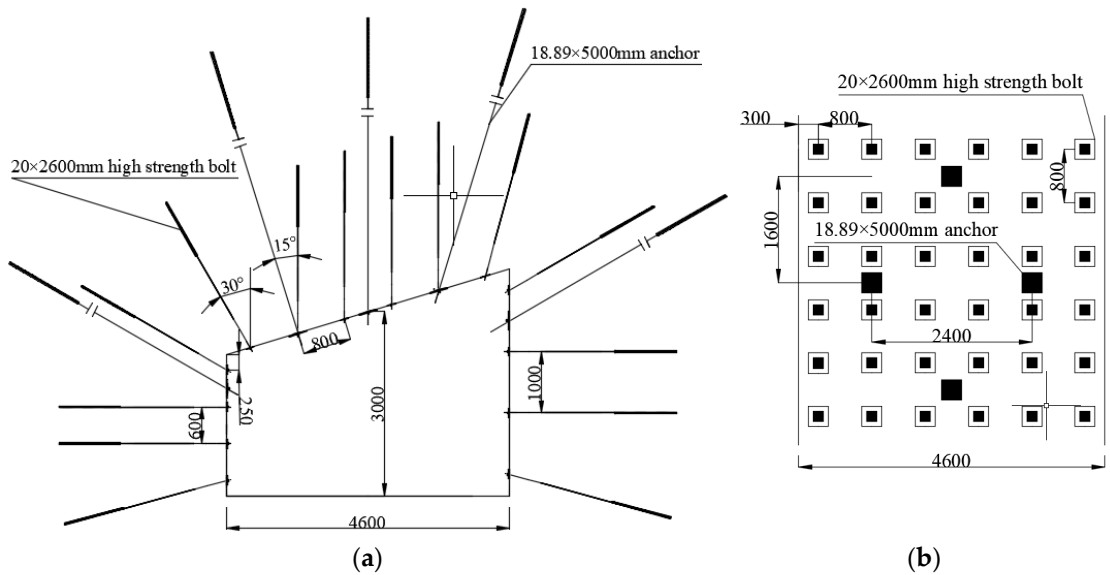

(a)          (b)

**Figure 10.** Support scheme. (**a**) Front view. (**b**) Top view of roof support.

### 5.2. Auxiliary Support System

The roadway roof is supported by Φ 18.89 mm × 5000 mm mining anchor cables. The upper parts of the two sides of the roadway are both equipped with a Φ 18.89 mm × 5000 mm anchor cable for reinforcement support. The installation preload is 100 kN, equipped with 300 × 300 × 12 mm high-strength arc trays, and the cable layout is given in Figure 10. The

anchor cables are matched with pressure-yield pipes. The pressure-yield distance is 26 mm, and the pressure-yield point is 220~250 kN (Figure 9b).

## 6. Engineering Experiment

An engineering experiment of the support scheme was conducted in the 4602 head roadway with the coal pillar width being 8 m. During excavation, mine pressure observation points are arranged along the roadway to monitor the roadway support quality, thereby monitoring the reasonableness of the selected roadway support parameters. These provide a basis for the optimization and adjustment of our roadway support scheme. Two monitoring stations were arranged in the 4602 head roadway, and the distance between each monitoring station was 50 m. The monitoring content of each measuring station included the displacement of the roadway surrounding rock and the stress on the anchor bolt (cable), and a specially assigned person monitored and collected the data regularly. The monitoring results were analyzed as follows.

### 6.1. Roadway Deformation Data and Analysis

Based on the data recorded at the observation points of the roadway surrounding the rock displacement, curves of the roadway displacement variation with working face distance were drawn (Figure 11). The roadway is basically stable when excavation advances to the observation point about 80 m away from the working face. Influenced by the gob= and coal-pillar-concentrated stress in the #1 coal seam as well as the coal pillars in the #2 coal seam, the coal pillar sides deform markedly (up to 82 mm) while the working face sides deform relatively less (up to 65 mm). When excavation advances to the observation point 80 m away from the working face, the roof subsidence curve stabilizes and becomes essentially linear, with a maximum roof subsidence of 95 mm.

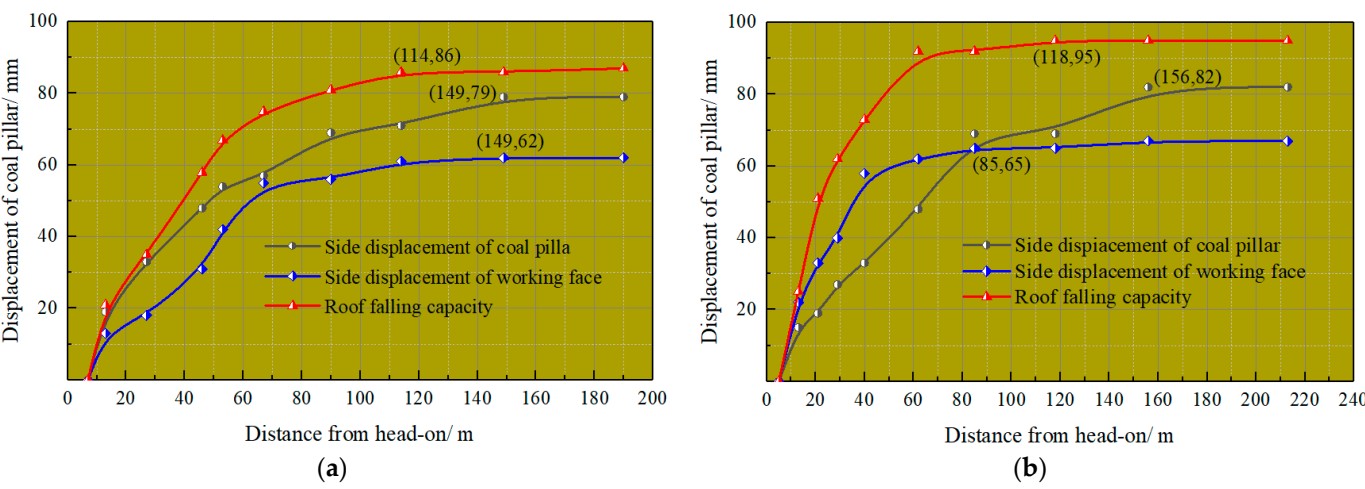

**Figure 11.** The curves of roadway displacement variation with working face distance. (**a**) Displacement of the #1 measuring station. (**b**) Displacement of the #2 measuring station.

According to the roadway displacement, the displacements of the roof and two sides are well controlled. The coal pillar sides deform quite markedly, and the displacement of the left side accounts for 56% of the deformation of the two sides.

### 6.2. Observed Data of Bolt Stress and Analysis

According to the data recorded at the two groups of bolt-stress observation points, the curves of bolt stress variation with working face distance were drawn (Figure 12). The bolt stress on the working face sides and roof increases linearly before the distance from the working face exceeds 80 m and tends to stabilize after the distance from the working face exceeds 80 m. Influenced by the coal pillars, the bolt stress on the coal pillar side is likely

to cause stress concentration, which leads to high bolt stress on the coal pillar side. The maximum bolt stress is 140 kN, and the maximum bolt stress of the working face side is 113 kN. The coal pillar side bolt is subject to a higher stress than the working face side bolt, and the maximum roof bolt stress is 122 kN. According to the data on anchor bolt stress, bolts are subject to high stress. The bolt stress on the coal pillar side and the roof is close to or even exceeds the load-yield point of the pressure-yield pipe. The pressure-yield device plays a role to make the anchor bolts bear the stress together and prevent the breakage of the local anchor bolt by excessive stress.

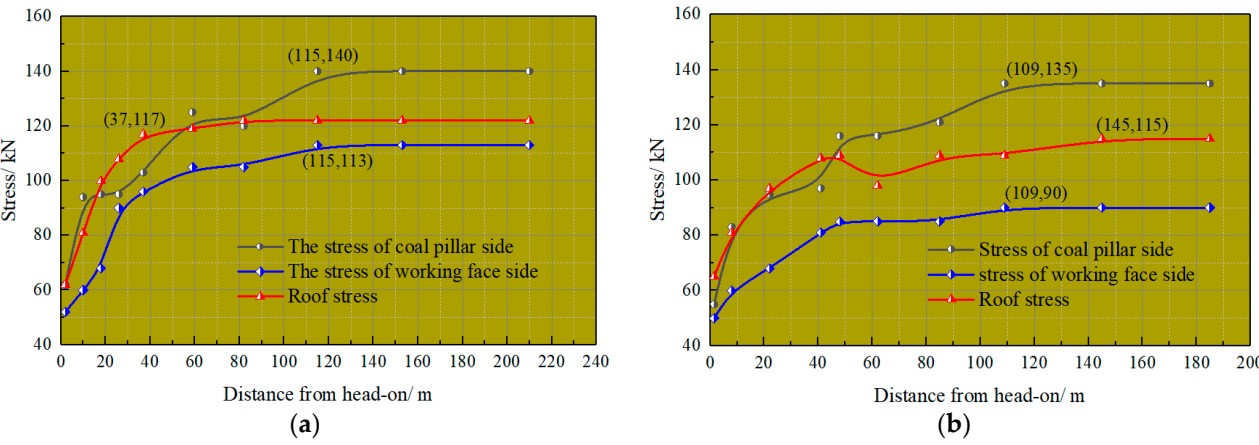

**Figure 12.** The curves of bolt stress variation with working face distance. (**a**) Load of the #1 measuring station. (**b**) Load of the #2 measuring station.

### 6.3. Observed Data of Anchor Stress and Analysis

According to the recorded data of the two groups of anchor cable stress, the curves of anchor cable stress variation with working face distance were drawn (Figure 13). After the installation of anchor cables, the stress of the anchor cable basically increases linearly and tends to stabilize when the observation point is approximately 100 m away from the working face, the maximum stress of the anchor cable being 210 kN, which is smaller than the cable-breaking strength of 400 kN. Cables can bear the stress together with anchor bolts and play the role of auxiliary strengthened support.

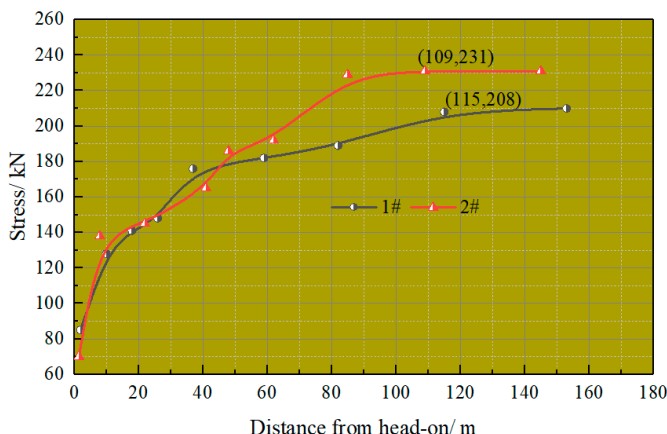

**Figure 13.** The curves of anchor cable stress variation with working face distance.

According to the above ground pressure observation and analysis, stable support of roadway excavation along the gob in the inclined coal seam is effectively realized by the pressure-anchor mesh-cable support scheme. The on-site support effect is illustrated in Figure 14.

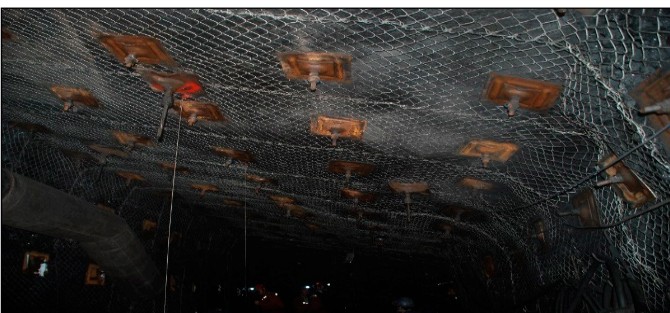

**Figure 14.** Effect drawing of on-site support.

*6.4. Potential Future Research*

In the future, we will continuously optimize the width of coal pillars and tunnel support schemes through theoretical and on-site testing methods. At the same time, we will conduct innovative research on anchor rod (cable) support technology to provide more support for safe, efficient, and green mining in mines.

**7. Conclusions**

(1) A theoretical model of short-distance coal seam floor failure and a mechanical model of inclined gob-side roadway were established. The influence depth of the residual coal pillars in the upper coal seam on the floor is 27 m, and the influence ranges of the stress concentration in coal pillars and the low-stress area of the inclined coal seam are 11 m and 12.6 m, respectively. With reference to the on-site production geological condition, the coal pillar width is preliminarily determined to be in the range of 6~8 m.

(2) Numerical simulation analysis reveals that when the width of coal pillars is smaller than 6 m, the internal stress of coal pillars is distributed in a single-peak shape, and stress is concentrated near the roadway. When the width is 8~14 m, the internal stress concentration position of coal pillars moves to the gob side, showing an asymmetric bimodal distribution of high stress on the gob side and low stress on the roadway side.

(3) Through numerical simulation calculation, the residual coal pillar has a certain impact on the mining of the #1 coal seam, and the influence range of the stress concentration is 10.5 m, which is almost consistent with the theoretical calculation of the concentrated influence range of the remaining coal pillar of the coal seam, which is 11 m, verifying the accuracy of the theoretical calculation. As the coal pillar width rises, the overall peak stress increases first and then decreases. When the coal pillar width reaches 8 m, the coal pillars have an overall strong bearing capacity.

(4) The gob-side entry driving in inclined short-distance coal seams should adopt the technology of pressure-yield anchor net cable support. The technology boasts good overall support, realizes the safety and stability of roadway support and provides a reference for mine roadway support under similar conditions.

**Author Contributions:** Conceptualization, F.H.; Methodology, W.Z.; Software, J.S.; Formal analysis, X.X.; Data curation, W.Z.; Writing—original draft, W.Z.; Writing—review & editing, D.W. and Y.W.; Supervision, F.H.; Project administration, F.H. All authors have read and agreed to the published version of the manuscript.

**Funding:** This work was supported by the National Natural Science Foundation of China (No.51974317), and the Fundamental Research Funds for the Central Universities (2022YJSNY09).

**Data Availability Statement:** The data used to support the findings of this study are included in the article.

**Conflicts of Interest:** The authors declare no conflict of interest.

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
