# Peer review of "Reasonable Coal Pillar Width and Surrounding Rock Control of Gob-Side Entry Driving in Inclined Short-Distance Coal Seams"

_applsci, doi:10.3390/app13116578_

Round 1
Reviewer 1 Report
Dear authors,
1. The study emphasizes the need for adequate pillar width to improve the stability and efficiency of mine production under complex conditions. However, it does not provide a comparison of the new proposed method with conventional methods, nor does it describe the limitations of existing methods. Therefore, it is difficult to assess the novelty and significance of the proposed pressure expansion technology without knowing the current state of the art.
2. The study does not include a detailed analysis of the economic feasibility of the proposed solution, which could affect its acceptance in the mining industry.
3. The introductory part of the paper is inadequate given the research on the subject. The summary and conclusion are also not good for such a research paper and should be improved. In particular, avoid using bullet points in these parts to prevent the paper from looking like a technical report.
4. Some parts of the paper include different font sizes in the text.
5. Since there are many papers on the current topic, you should make it clearer what the main contribution of your paper is to science in order to close the novelty gap of the manuscript.
6. The main problem is that the study has a similarity rate of 36% according to the iThenticate report, which prevents a detailed review of your results. Therefore, you need to reduce it in the revision.
Best regards.
Author Response
Response to Reviewer Comments and Detailed Explanation of Revisions
For the manuscript titled,
Reasonable coal Pillar Width and Surrounding Rock Control of Gob-side Entry Driving in Inclined Short-distance Coal Seams
Manuscript Number,
applsci-2365874
Submitted to “Applied Sciences”
Comment from reviewer 1:
Point 1.
The study emphasizes the need for adequate pillar width to improve the stability and efficiency of mine production under complex conditions. However, it does not provide a comparison of the new proposed method with conventional methods, nor does it describe the limitations of existing methods. Therefore, it is difficult to assess the novelty and significance of the proposed pressure expansion technology without knowing the current state of the art.
Reply:
Thank you very much for your careful review and valuable comments, according to your comments, our manuscript has been checked carefully and revised point to point, the revised portions are marked in red in the manuscript.
Point 2.
The study does not include a detailed analysis of the economic feasibility of the proposed solution, which could affect its acceptance in the mining industry.
Reply:
Our study does not include a detailed analysis of the economic feasibility of the proposed solution, The pressure-yield anchor net cable support technology is not significantly different from the original support technology, and the overall construction process and process are basically similar. The gob-side entry driving in inclined short-distance coal seams adopts the technology of pressure-yield anchor net cable support, and realizes the safety and stability of roadway support.
Point 3.
The introductory part of the paper is inadequate given the research on the subject. The summary and conclusion are also not good for such a research paper and should be improved. In particular, avoid using bullet points in these parts to prevent the paper from looking like a technical report.
Reply:
We have been modified in the manuscript.
Point 4.
Some parts of the paper include different font sizes in the text.
Reply:
We have been modified in the manuscript.
Point 5.
Since there are many papers on the current topic, you should make it clearer what the main contribution of your paper is to science in order to close the novelty gap of the manuscript.
Reply:
We have been modified to the manuscript.
Point 6.
The main problem is that the study has a similarity rate of 36% according to the iThenticate report, which prevents a detailed review of your results. Therefore, you need to reduce it in the revision.
Reply:
We have been modified to the manuscript.

Reviewer 2 Report
Point 1.
The paper explores a way to maintain stability and safety in coal mine production by determining a reasonable coal pillar width in inclined short-distance coal seams. Through on-site investigation, theoretical analysis, numerical simulation and engineering test, the study found that a coal pillar width of 8m is suitable and proposed a new pressure-yield support technology which was tested on-site and had a positive effect. Finally, the researchers proposed a new pressure-yield support technology, which was confirmed to have a significant impact on roadway control through on-site application. This research is important because it provides a solution to a common problem in the mining industry: instability and deformation of roadways during gob-side entry driving. So, the study explores the optimal width of coal pillars and proposes a new pressure-yield support technology to improve roadway stability and efficiency in coal mines. By considering geological conditions and conducting on-site investigations, numerical simulations, and engineering tests, the research provides practical guidance for mine operators to improve safety and productivity in their operations.
Point 2.
The abstract section sounds unclear. The abstract should follow the MDPI style of structured abstracts: Background (place the question addressed in a broad context and highlight the purpose of the study); Methods (describe briefly the main methods); Results (summarize the article's main findings); Conclusion (indicate the main conclusions or interpretations).
Point 3.
Please consider to rise following information in you research:
What are the common challenges associated with gob-side entry driving in inclined short-distance coal seams, and how does the pillar width affect surrounding rock control?
What is the optimal pillar width for gob-side entry driving in inclined short-distance coal seams, and what factors should be considered in determining the width?
What are the key techniques for ensuring effective surrounding rock control during gob-side entry driving in inclined short-distance coal seams?
How do different geological and mining conditions affect the choice of pillar width and surrounding rock control techniques in gob-side entry driving in inclined short-distance coal seams?
What are some successful case studies of gob-side entry driving in inclined short-distance coal seams, and what lessons can be learned from these experiences in terms of pillar width and surrounding rock control?
Point 4.
In the Introduction section, an enhanced literature review is required. For this study, the authors have used only 18 literature sources. It is insufficient for such type of research. Moreover almost all references comes from China.
Point 5.
Providing a contextual explanation of why this study is significant would be highly beneficial for the readers.
Point 6.
Could you please elaborate on the constraints or drawbacks of your research?
Point 7. In case to add more information (literature review according to Point 4) please consider the suggested research (comes from, Ukraine, Kazakhstan and Poland) in your paper when enhancing the introduction section. I believe they are worth considering in your paper.
Krykovskyi, O.; Krykovska, V.; Skipochka, S. Interaction of rock-bolt supports while weak rock reinforcing by means of injection rock bolts. Min. Miner. Depos. 2021, 15(4), 8-14. https://doi.org/10.33271/mining15.04.008
Dyczko, A.; KamiÅ„ski, P.; Jarosz, J.; Rak, Z.; Jasiulek, D.; Sinka, T. Monitoring of Roof Bolting as an Element of the Project of the Introduction of Roof Bolting in Polish Coal Mines—Case Study. Energies, 2021, 15, 95. https://doi.org/10.3390/en15010095
Vu, T.T. Solutions to prevent face spall and roof falling in fully mechanized longwall at underground mines, Vietnam. Min. Miner. Depos. 2022, 16, 127-134. https://doi.org/10.33271/mining16.01.127
Point 8.
Stress monitoring curve (Figure 7 b) need to be explained in the text.
Point 9.
Lines 262-266. Please provide more detail.
Point 10.
What about the influence of leaving rock underground after coal mining on pillar stability? Can such actions make influence damage evolution law? Please consider mentioned below research in your study as it can affect further research direction.
Smoliński, A.; Malashkevych, D.; Petlovanyi, M.; Rysbekov, K.; Lozynskyi, V.; Sai, K. Research into Impact of Leaving Waste Rocks in the Mined-Out Space on the Geomechanical State of the Rock Mass Surrounding the Longwall Face. Energies, 2022, 15, 9522. https://doi.org/10.3390/en15249522
Malashkevych, D.; Petlovanyi, M.; Sai, K.; Zubko, S. Research into the coal quality with a new selective mining technology of the waste rock accumulation in the mined-out area. Min. Miner. Depos. 2022, 16, 103-114. https://doi.org/10.33271/mining16.04.103
Point 11.
The references are not in the correct format of MDPI. Please make the necessary corrections.
Point 12.
The authors should include a brief summary of potential future research as providing a description of the importance of an issue or study helps to establish its relevance and potential impact. It also helps to provide context and persuade readers to engage with the research.
In general, the presented article leaves a positive impression and, after eliminating these comments and taking into account the recommendations made, it will be recommended for publication in the journal “Applied Sciences”.
Author Response
Response to Reviewer Comments and Detailed Explanation of Revisions
For the manuscript titled,
Reasonable coal Pillar Width and Surrounding Rock Control of Gob-side Entry Driving in Inclined Short-distance Coal Seams
Manuscript Number,
applsci-2365874
Submitted to “Applied Sciences”
Comment from reviewer 2:
Point 1.
The paper explores a way to maintain stability and safety in coal mine production by determining a reasonable coal pillar width in inclined short-distance coal seams. Through on-site investigation, theoretical analysis, numerical simulation and engineering test, the study found that a coal pillar width of 8m is suitable and proposed a new pressure-yield support technology which was tested on-site and had a positive effect. Finally, the researchers proposed a new pressure-yield support technology, which was confirmed to have a significant impact on roadway control through on-site application. This research is important because it provides a solution to a common problem in the mining industry: instability and deformation of roadways during gob-side entry driving. So, the study explores the optimal width of coal pillars and proposes a new pressure-yield support technology to improve roadway stability and efficiency in coal mines. By considering geological conditions and conducting on-site investigations, numerical simulations, and engineering tests, the research provides practical guidance for mine operators to improve safety and productivity in their operations.
Reply:
Thank you very much for your careful review and valuable comments, according to your comments, our manuscript has been checked carefully and revised point to point, the revised portions are marked in red in the manuscript.
Point 2.
The abstract section sounds unclear. The abstract should follow the MDPI style of structured abstracts: Background (place the question addressed in a broad context and highlight the purpose of the study); Methods (describe briefly the main methods); Results (summarize the article's main findings); Conclusion (indicate the main conclusions or interpretations).
Reply:
We have been modified in the manuscript.
During gob-side entry driving under complex conditions in inclined short-distance coal seams, the roadway loses stability and deforms seriously, which affects the safety and efficiency of mine production. In this study, a reasonable coal pillar width was explored by means of on-site inves-tigation, theoretical analysis, numerical simulation and engineering test. The following research results were obtained. (1) In selecting a reasonable coal pillar width, the influences of the position of residual coal pillars, stratum spacing, main roof breakage, roadway section in the upper coal seam should be considered, and established mechanical models of inclined gob-side roadways, the maximum floor failure depth is 27m and the concentrated influence range of the 1# coal pillars is 11 m. (2) The stress states of coal pillars with different widths were analyzed by numerical simula-tion, the coal pillar width increases, the peak value of stress increases first and then decreases, according to the site geological conditions, the coal pillar width is finally determined to be 8 m, whose results are basically consistent with the theoretical calculation results. (3) A new pres-sure-yield support technology was proposed, and its on-site application confirmed its notable roadway control effect. Our research can provide certain theoretical support for the control of roadway surrounding rock under similar engineering background conditions.
Point 3.
Please consider to rise following information in you research:
What are the common challenges associated with gob-side entry driving in inclined short-distance coal seams, and how does the pillar width affect surrounding rock control?
What is the optimal pillar width for gob-side entry driving in inclined short-distance coal seams, and what factors should be considered in determining the width?
What are the key techniques for ensuring effective surrounding rock control during gob-side entry driving in inclined short-distance coal seams?
How do different geological and mining conditions affect the choice of pillar width and surrounding rock control techniques in gob-side entry driving in inclined short-distance coal seams?
What are some successful case studies of gob-side entry driving in inclined short-distance coal seams, and what lessons can be learned from these experiences in terms of pillar width and surrounding rock control?
Reply:
We have added it in the manuscript.
Point 4.
In the Introduction section, an enhanced literature review is required. For this study, the authors have used only 18 literature sources. It is insufficient for such type of research. Moreover almost all references comes from China.
Reply:
We have been modified in the manuscript.
Point 5.
Providing a contextual explanation of why this study is significant would be highly beneficial for the readers.
Reply:
In originally adopted gob-side entry driving, the roadway has a rectangular section with a net width of 4,000 mm and a net height of 3,000 mm. The roadway roof and sides adopt Ф 20 mm × 2,400 mm high-strength bolt support with a row spacing of 800 mm × 800 mm. The anchor cables are Ф 18.9 mm × 5,000 mm mining anchor cables; each row has 3 cables with a spacing of 1,200 mm, and the row spacing is 1,600 mm. After the above support scheme is adopted, the roadway surrounding rock deforms seriously during recovery, especially the gob-side roof and sides of the roadway where the bolts move outward with the deformation of surrounding rock. At the same time, local floor heave occurs in the roadway. The recovery requires repeated side expansion and floor reshaping. If it gets too serious, grouting and re-building metal U-shaped steel shed support are needed after the repair. According to the above road-way deformation characteristics of gob-side entry driving, coal pillar width and support parameters require further optimization.
Point 6.
Could you please elaborate on the constraints or drawbacks of your research?
Reply:
The physical and mechanical parameters of rock layers are obtained through experiments, but there is a certain error between them and the actual situation. Therefore, it has a certain impact on the numerical simulation results.
Point 7.
In case to add more information (literature review according to Point 4) please consider the suggested research (comes from, Ukraine, Kazakhstan and Poland) in your paper when enhancing the introduction section. I believe they are worth considering in your paper.
Krykovskyi, O.; Krykovska, V.; Skipochka, S. Interaction of rock-bolt supports while weak rock reinforcing by means of injection rock bolts. Min. Miner. Depos. 2021, 15(4), 8-14. https://doi.org/10.33271/mining15.04.008
Dyczko, A.; KamiÅ„ski, P.; Jarosz, J.; Rak, Z.; Jasiulek, D.; Sinka, T. Monitoring of Roof Bolting as an Element of the Project of the Introduction of Roof Bolting in Polish Coal Mines—Case Study. Energies, 2021, 15, 95. https://doi.org/10.3390/en15010095
Vu, T.T. Solutions to prevent face spall and roof falling in fully mechanized longwall at underground mines, Vietnam. Min. Miner. Depos. 2022, 16, 127-134. https://doi.org/10.33271/mining16.01.127
Reply:
We have added it in the manuscript.
Point 8.
Stress monitoring curve (Figure 7 b) need to be explained in the text.
Reply:
We have been modified in the manuscript.
Point 10.
What about the influence of leaving rock underground after coal mining on pillar stability? Can such actions make influence damage evolution law? Please consider mentioned below research in your study as it can affect further research direction.
Smoliński, A.; Malashkevych, D.; Petlovanyi, M.; Rysbekov, K.; Lozynskyi, V.; Sai, K. Research into Impact of Leaving Waste Rocks in the Mined-Out Space on the Geomechanical State of the Rock Mass Surrounding the Longwall Face. Energies, 2022, 15, 9522. https://doi.org/10.3390/en15249522.
Malashkevych, D.; Petlovanyi, M.; Sai, K.; Zubko, S. Research into the coal quality with a new selective mining technology of the waste rock accumulation in the mined-out area. Min. Miner. Depos. 2022, 16, 103-114. https://doi.org/10.33271/mining16.04.103.
Reply:
We have added it in the manuscript.
Point 11.
The references are not in the correct format of MDPI. Please make the necessary corrections.
Reply:
We have been modified in the manuscript.
Point 12.
The authors should include a brief summary of potential future research as providing a description of the importance of an issue or study helps to establish its relevance and potential impact. It also helps to provide context and persuade readers to engage with the research.
Reply:
We have been modified in the manuscript.

Reviewer 3 Report
Underground mining of hard coal deposits using a longwall method requires good recognition of geological conditions, in particular in the conditions of the impact of old goafs and mining edges. The presented research issue is interesting in terms of science and technology and can be helpful at the design stage of selecting the support for the tailgate and maingate excavations. The use of an yielding support in industrial conditions for the width of the calculated pillar is an additional advantage and has a utilitarian aspect. Below are some comments and suggestions:
1. In the introduction, some information should be added regarding rock mass modeling using physical models that take into account the inclination of the deposit;
2. The article should use a space between the number and the unit (lines 69, 72, 183-185, ). In addition, please use one type for decimal numbers - (line 85…);
3. In the subsection 2.1 it should be described how the post-mining space was liquidated - whether it was a collapse of roof rocks or backfilling;
4. In the subsection 2.2, it should be written how the bolts and anchors were fixed along the entire length or in sections, what type of binder was used for their embedment – resin cardridges or cement binder;
5. Line 90, sentence: "…..local floor heave occurs in the roadway…." It should be specified the size of the uplift either as a percentage of the excavation cross-section or in mm;
6. Figure 3a - the word "rip" should it be "rib";
7. Line 109, the sign "<" in the OFE, EOF equation should be corrected;
8. Figure 5b - the letter "h" coincides with another letter, it should be corrected;
9. Line 178, x - influence coefficient, it should be written in what limits is this coefficient for horizontal and inclined coal seams - is there any difference;
10. In the fourth chapter, it should be written which: numerical program was used in the research, whether the models included strength and deformation parameters for the post-mining space;
11. Chapter five, line 292, sentence "....260 N·m pre-load..." – if it is a unit of Nm, it should be written about the tightening torque;
12. Line 295, the statement "pressure" is quite misleading because the unit of pressure is MPa, if we are talking about kN it should be load;
13. Explain in section 5.1 why bolt and cable anchor support with yielding pipe were used. In addition, please add information that flexible support is commonly used in conditions of high rock mass deformation (doi.org/10.3390/en15072574);
14. Figures 11a-b, description on the vertical axis - the word amount is not needed because the displacement can be measured; moreover, it is better to use displacement instead of deformation, especially that the description includes the characteristics for sidewalls and roof;
15. Figures 12 and 13, vertical axis, unit for pressure to MPa, should be load (kN) - this should be corrected;
16. In chapter six, it should be mentioned that the loading of headings in inclined deposits requires the selection of the appropriate type of mining support, which should be correlated with the convergence of the heading;
17. The conclusions include the results of numerical tests for the pillar width and present recommendations for the mine, therefore they are sufficient.
Author Response
Response to Reviewer Comments and Detailed Explanation of Revisions
For the manuscript titled,
Reasonable coal Pillar Width and Surrounding Rock Control of Gob-side Entry Driving in Inclined Short-distance Coal Seams
Manuscript Number,
applsci-2365874
Submitted to “Applied Sciences”
Comment from reviewer 3:
Underground mining of hard coal deposits using a longwall method requires good recognition of geological conditions, in particular in the conditions of the impact of old goafs and mining edges. The presented research issue is interesting in terms of science and technology and can be helpful at the design stage of selecting the support for the tailgate and maingate excavations. The use of an yielding support in industrial conditions for the width of the calculated pillar is an additional advantage and has a utilitarian aspect. Below are some comments and suggestions:
Reply:
Thank you very much for your careful review and valuable comments, according to your comments, our manuscript has been checked carefully and revised point to point, the revised portions are marked in red in the manuscript. The responses to the comments are as follows.
Point 1.
In the introduction, some information should be added regarding rock mass modeling using physical models that take into account the inclination of the deposit;
Reply:
We have added it in the manuscript.
Point 2.
The article should use a space between the number and the unit (lines 69, 72, 183-185). In addition, please use one type for decimal numbers - (line 85…);
Reply:
We have modified it in the manuscript.
Point 3.
In the subsection 2.1 it should be described how the post-mining space was liquidated - whether it was a collapse of roof rocks or backfilling;
Reply:
Collapse of roof rocks.
Point 4.
In the subsection 2.2, it should be written how the bolts and anchors were fixed along the entire length or in sections, what type of binder was used for their embedment – resin cardridges or cement binder; Reply:
We have added it in the manuscript.
Point 5.
- Line 90, sentence: "…..local floor heave occurs in the roadway…." It should be specified the size of the uplift either as a percentage of the excavation cross-section or in mm;
Reply:
We have added it in the manuscript.
Point 6.
- Figure 3a - the word "rip" should it be "rib";
Reply:
We have modified it in the manuscript.
Point 7.
Line 109, the sign "<" in the OFE, EOF equation should be corrected;
Reply:
We have modified it in the manuscript.
Point 8.
Figure 5b - the letter "h" coincides with another letter, it should be corrected;
Reply:
We have been modified in the manuscript.
Point 9.
Line 178, x - influence coefficient, it should be written in what limits is this coefficient for horizontal and inclined coal seams - is there any difference;
Reply:
No difference.
Point 10.
In the fourth chapter, it should be written which: numerical program was used in the research, whether the models included strength and deformation parameters for the post-mining space;
Reply:
No treatment measures have been taken in the post-mining space. In future research, we will use filling and other methods to treat the post-mining space.
Point 11.
Chapter five, line 292, sentence "....260 N·m pre-load..." – if it is a unit of Nm, it should be written about the tightening torque;
Reply:
We have been modified in the manuscript.
Point 12.
Line 295, the statement "pressure" is quite misleading because the unit of pressure is MPa, if we are talking about kN it should be load;
Reply:
We have been modified in the manuscript.
Point 13.
Explain in section 5.1 why bolt and cable anchor support with yielding pipe were used. In addition, please add information that flexible support is commonly used in conditions of high rock mass deformation (doi.org/10.3390/en15072574);
Reply:
We have added it in the manuscript.
Point 14.
Figures 11a-b, description on the vertical axis - the word amount is not needed because the displacement can be measured; moreover, it is better to use displacement instead of deformation, especially that the description includes the characteristics for sidewalls and roof;
Reply:
We have added it in the manuscript.
Point 15.
Figures 12 and 13, vertical axis, unit for pressure to MPa, should be load (kN) - this should be corrected;
Reply:
We have added it in the manuscript.
Point 16.
In chapter six, it should be mentioned that the loading of headings in inclined deposits requires the selection of the appropriate type of mining support, which should be correlated with the convergence of the heading;
Reply:
We have added it in the manuscript.
Point 17.
The conclusions include the results of numerical tests for the pillar width and present recommendations for the mine, therefore they are sufficient.
Reply:
Thank you very much for your careful review and valuable comments, according to your comments, our manuscript has been checked carefully and revised point to point.

Round 2
Reviewer 1 Report
Dear authors,
It would be nice if you could respond to the comments in more detail to recommend your work to readers.
Author Response
Our research is based on the engineering background of inclined close range coal seams, establishing a mechanical model of residual coal pillars in close range coal seams, analyzing the influence range of residual coal pillars, and analyzing the range of stress field in the lower coal seam, and preliminarily determining the width of coal pillars; Based on numerical simulation analysis, the final width of the coal pillar was determined to be 8m. A new pressure-yield support technology was proposed, and its on-site application confirmed its notable roadway control effect.
Reviewer 2 Report
Dear authors,
I am wondering if the final version of the paper was resubmitted. Why? Because I am satisfied with the accuracy of the revision according to my Points 1-8, but 9-12 not.
Point 9. - NO RESPONSE
Point 10. - Authors replied (We have added it in the manuscript), but I can not find it in the paper.
Point 11. - Authors replied (We have been modified in the manuscript), but the references are still not prepared in MDPI format
Point 12. - Authors replied (We have been modified in the manuscript), but a brief summary of potential future research (before conclusions) is not provided.
Please carefully revise the paper once more.
Author Response
Response to Reviewer Comments and Detailed Explanation of Revisions
For the manuscript titled,
Reasonable coal Pillar Width and Surrounding Rock Control of Gob-side Entry Driving in Inclined Short-distance Coal Seams
Manuscript Number,
applsci-2365874
Submitted to “Applied Sciences”
Comment from reviewer 2:
Point 9.
Lines 262-266. Please provide more detail.
Reply:
We have been modified in the manuscript.
Point 10.
What about the influence of leaving rock underground after coal mining on pillar stability? Can such actions make influence damage evolution law? Please consider mentioned below research in your study as it can affect further research direction.
Smoliński, A.; Malashkevych, D.; Petlovanyi, M.; Rysbekov, K.; Lozynskyi, V.; Sai, K. Research into Impact of Leaving Waste Rocks in the Mined-Out Space on the Geomechanical State of the Rock Mass Surrounding the Longwall Face. Energies, 2022, 15, 9522. https://doi.org/10.3390/en15249522.
Malashkevych, D.; Petlovanyi, M.; Sai, K.; Zubko, S. Research into the coal quality with a new selective mining technology of the waste rock accumulation in the mined-out area. Min. Miner. Depos. 2022, 16, 103-114. https://doi.org/10.33271/mining16.04.103.
Reply:
We have added the above two references to our manuscript (lines 59-62).
Point 11.
The references are not in the correct format of MDPI. Please make the necessary corrections.
Reply:
We have been modified in the manuscript.
Point 12.
The authors should include a brief summary of potential future research as providing a description of the importance of an issue or study helps to establish its relevance and potential impact. It also helps to provide context and persuade readers to engage with the research.
Reply:
We have added to our manuscript (lines 369-373).

Reviewer 3 Report
The article has been well improved.
Author Response
Thank you very much for your careful review and valuable comments.

Round 3
Reviewer 1 Report
Dear authors,
I appreciate your efforts to improve your paper for possible publication in the journal. Although I did not gain any new ideas from the paper, I think that readers of the journal who are interested in the subject can get some benefits from the paper.
Best regards.
Reviewer 2 Report
Perfect. Your paper is a good contribution to mining science.
Wish all the best in your research activities.
Author Response

(The authors gave the same response as above.)
